# Neutralising Antibodies against Enterovirus and Parechovirus in IVIG Reflect General Circulation: A Tool for Sero-Surveillance

**DOI:** 10.3390/v13061028

**Published:** 2021-05-29

**Authors:** Karen Couderé, Karlijn van der Straten, Lieke Brouwer, Gerrit Koen, Hetty van Eijk, Dasja Pajkrt, Jean-Luc Murk, Katja C. Wolthers

**Affiliations:** 1Department of Medical Microbiology Microvida, Elisabeth-Tweesteden Ziekenhuis, 5022 GC Tilburg, The Netherlands; j.murk@etz.nl; 2Department of Medical Microbiology, Amsterdam University Medical Centres, Locatie AMC, 1105 AZ Amsterdam, The Netherlands; k.vanderstraten@amsterdamumc.nl (K.v.d.S.); liekebrouwer@hotmail.com (L.B.); g.koen@amsterdamumc.nl (G.K.); h.w.vaneijk@amsterdamumc.nl (H.v.E.); k.c.wolthers@amsterdamumc.nl (K.C.W.); 3Emma Children’s Hospital, Department of Pediatric Infectious Diseases, Amsterdam University Medical Centres, Location AMC, 1105 AZ Amsterdam, The Netherlands; d.pajkrt@amsterdamumc.nl

**Keywords:** *Enterovirus*, *Parechovirus*, surveillance, sero-surveillance, virus neutralisation assay

## Abstract

Non-polio enteroviruses (NPEV) and parechoviruses (PeV) are widespread pathogens that cause significant morbidity. Surveillance is based on culturing or genotyping of virus strains found in clinical samples. Sero-surveillance, by measuring neutralising antibodies (nAb) through virus neutralisation assays (VNA), could provide additional information as it offers a more comprehensive overview of exposure to circulating types in the general population. In our study we evaluated Intravenous immunoglobulins (IVIG) to generate sero-surveillance data. We performed VNA of nineteen NPEV and PeV with Dutch IVIG batches from two different time points (2010 and 2017) and an IVIG batch from Vietnam (2011). We compared our findings with geno- and sero-surveillance data and evaluated changes over time and between the two countries. Our findings show a good correlation with what is known from geno-surveillance data. The highest nAb titres were found against strains from *Enterovirus B*, while we did not observe nAb titres against strains belonging to *Enterovirus C*. In conclusion, we demonstrated that sero-surveillance by means of IVIG can be used to obtain insight into circulation of EV and PeV genotypes. This is of particular interest for public health, to evaluate changes over time and population susceptibility to emerging genotypes.

## 1. Introduction

Numerous enteroviruses (EV) and parechoviruses (PeV) have recently caused large outbreaks, including EV-A71, EV-D68 and PeV-3 [1,2]. These viruses belong to the *Picornaviridae* family, one of the largest RNA virus families. Human EV and PeV cause a wide range of disease from mild upper respiratory infections to more serious infections including hand–foot–mouth disease (HFMD), myocarditis, neonatal sepsis and neurological disease [3]. They are the most common causative pathogen of viral meningitis [4,5] and some types are known to cause (meningo-) encephalitis, brainstem encephalitis and acute flaccid myelitis/paralysis (AFM/AFP) causing important morbidity and mortality worldwide. Human EV contain more than 100 different types and are subdivided into four species; *Enterovirus A* to *Enterovirus D*, which include coxsackie (CV) A- and B-, echo- and numbered enteroviruses (EV). Human PeV, of which PeV-1 and PeV-2 were originally misclassified as E22 and E23, belong to *Parechovirus A* species, which currently contains 19 types: PeV-A1 up to PeV-A19. Classification of EV and PeV is based on molecular typing which is performed by partial sequencing of the viral capsid protein 1 (VP1) region. These genotypes correlate largely with previously defined serotypes as this region contains most of the neutralising epitopes [2]. The term non-polio enteroviruses (NPEV) refers to all enterovirus types except for the three poliovirus types. Despite the serious morbidity NPEV and PeV can cause, many countries do not have systematic surveillance in place and knowledge on circulating types is mainly based on genotyping through VP1 sequencing of EV and PeV found in clinical samples. Although it provides an important insight in the current types causing serious disease which require hospitalization, a disadvantage of this method is that it does not represent all circulating virus types in the population. Sero-surveillance, by detecting neutralising antibodies (nAb) against specific virus types through Virus Neutralisation Assays (VNA), reflects the magnitude at which a population has been in contact with certain virus types, including those causing mild or no disease and emerging subtypes [6]. This information, together with genotyping data, could help to predict circulation patterns [7]. Moreover, during outbreaks, it has the potential to provide information on the level of protection of a population against the outbreak strain. Sero-surveillance has several challenges however, like requiring access to large representative serum banks. Intravenous immunoglobulins (IVIG) provide the unique opportunity to investigate a representative sample from the general population as it contains plasma from at least 1000 healthy adult donors from different age groups [6,8]. In our study we evaluated IVIG as a specimen, representative of the population, to generate sero-surveillance data for the most commonly detected NPEV and PeV in the Netherlands. We analysed changes over time by comparing two Dutch IVIG batches from 2010 and 2017, and differences between geographical regions by comparing with a 2011 IVIG batch from Vietnam. The results were related to geno- and sero-surveillance data from the literature.

## 2. Materials and Methods

### 2.1. IVIG Selection

Three IVIG batches were selected; two from the Netherlands and one from Vietnam. The Dutch IVIG batches (Nanogam, Sanquin, the Netherlands) were fabricated in 2010 and 2017. The Vietnamese IVIG batch (Green Cross Corporation, Pymepharco, Vietnam) was fabricated in 2011 and kindly provided by R. van Doorn from the Ho Chi Minh Hospital for Tropical Diseases. The protein concentrations from the Dutch 2010, 2017 and Vietnamese 2011 IVIG batches were, respectively 49.9 mg/mL, 50.6 mg/mL and 55.1 mg/mL. As a positive control, we used PeV-A1 (Harris B) as we had previously detected high nAb titres against this strain in Dutch IVIG [9]. The positive control was neutralised by all three IVIG batches.

### 2.2. Selection of NPEV and PeV strains

The selection of prevalent circulating types was based on VIRO-TypeNed data. In the Netherlands this web-based platform is used for surveillance of EV and PeV with collection of demographic, clinical and sequence-typing data from laboratories countrywide on a voluntary basis [10]. We defined a highly prevalent type as being listed more than once in the top five most prevalent viruses circulating between 2011 and 2017 (Table 1). This resulted in the inclusion of the following twelve virus types: CVA6, E6, E9, E11, E18, E25, E30, CVB5, CVA9, PeV-A1, PeV-A3, PeV-A4. Additionally, three types were included based on association with disease: one EV-D68 strain, two EV-A71 strains and one CVA16 strain. Additionally, four types from *Enterovirus C* species, which rarely occur in the VIRO-TypeNed surveillance data, were included: CVA11, CVA13, CVA20 and CVA24. For each type, we selected the most recent isolated clinical strain from the Laboratory of Medical Microbiology Amsterdam University Medical Centres (Amsterdam UMC) location AMC, to match the current circulating strains. CVA11 was kindly provided by the Leiden University Medical Centre (LUMC). CVA13, CVA20, CVA24 and the EV-D68 strain were kindly provided by the National Institute of Public Health and Environment (RIVM). The EV-A71C4 strain from Japan was kindly provided by K. Mizuta from the Yamagata Prefectural Institute of Public Health. Characteristics of the selected strains can be found in Appendix A.

### 2.3. Cell Line Selection

All strains were cultured on six different cell lines: Human Colorectal Adenocarcinoma (HT29), African Green Monkey Kidney (Vero), Rhabdomyosarcoma (RD), Rhesus Monkey Kidney Epithelial (LLC-MK2), Human Embryo Lung (HEL) and Human Lung Adenocarcinoma (A549). Cells were cultured at 37 °C 5% carbon dioxide (CO_2_) in Eagles minimum essential medium (EMEM, Lonza, Switzerland) supplemented with 8% foetal bovine serum (FBS, Sigma-Aldrich, United States), 100 IU/mL penicillin and 100 μg/mL streptomycin. We selected the most appropriate cell line for each viral strain, that is with a cytopathological effect ≥50% within 7 days, to perform the serum neutralisation assay. HT cells were selected for most viruses (9/19), followed by RD (6/19), Hel (3/19) and LLC-MK2 (1/19). CVA24 did not grow on any of the cell lines and therefore had to be excluded from further experiments.

### 2.4. Dilution Neutralisation Assay

Neutralising antibody (nAb) titres of IVIG batches against the selected viruses were determined by using a virus neutralisation assay. An in-duplicate 2-fold serial dilution of the IVIG batches were incubated with an equal volume of 100 μL 50% tissue culture infective dose (TCID_50_) of virus at 37 °C in 5% CO_2_ for one hour. PeV-3 and virus strains that tested negative in the first neutralisation assay (EV-A71C1, CVA16, E18, CVA11, CVA13 and CVA20) were pre-treated with chloroform to overcome virus aggregation and optimize neutralisation [11,12]. HT29, RD, Hel, Vero or LLC-MK2 cells in EMEM supplemented with 8% FBS were subsequently added and incubated for 7 days at 37 °C 5% CO_2_. We determined nAb titres based on the cytopathologic effect using the Reed and Muench method. We reported nAb titres as the reciprocal titres of serum dilutions exhibiting 50% neutralisation. Titres of ≥1:16 and ≥1:32 were defined as seropositive and protective, respectively. We divided nAb titres based on their level: absence of titres (<1:16), low titres (1:16–1:128), medium titres (1:129–1:511) and high titres (1:512–1:2048) [13,14].

### 2.5. Statistical Analysis

Statistics were performed in R (R core team 2020). We used the Kruskal–Wallis test to compare nAb titres between the different species and the Mann–Whitney–Wilcoxon test to compare nAb titres between IVIG batches for the different species. *p* < 0.05 was defined as significant. A titre difference of more than 2-fold between nAb against two individual types was deemed relevant, while a difference of 2-fold or less was determined as inter-assay variability.

## 3. Results

A total of 19 virus types were included: 3 *Parechovirus A*, 4 *Enterovirus A*, 8 *Enterovirus B*, 3 *Enterovirus C* and 1 *Enterovirus D* type. The nAb titres from the different IVIG batches against the selected viral strains are presented in Figure 1 and Appendix A.

### 3.1. Neutralising Ab Titres Against EV and PeV in Dutch 2017 IVIG

In the Dutch IVIG from 2017, the geometric mean nAb titres (GMT) were medium against the species *Enterovirus A, Enterovirus B* and *Parechovirus A* (1:241, 1:373 and 1:342, respectively) (Figure 1a). The nAb titres between types from *Enterovirus A*, *Enterovirus B* and *Parechovirus A* did not differ significantly (*p* = 0.275). No nAb titres (<1:4) were found against any of the types from *Enterovirus C* species, including after chloroform treatment. The 2017 IVIG contained nAb titres against the three virus types belonging to *Enterovirus A*. A high nAb titre (1:512) was found against CVA16 and medium nAb titres were found against CVA6 and EV-A71. The 2017 IVIG contained nAb titres against all *Enterovirus B* types tested. It showed low nAb titres against E30 (1:97), despite the high frequency of detection according to VIRO-Typened (Table 1). Neutralising Ab titres against other types of EV-B species ranged from medium (E6, E11, E25, E18) to high (E9, CVB5, CVA9). A nAb titre of 1:128 was found against EV-D68. PeV-A3 and PeV-A4 showed medium and high nAb titres, respectively. PeV-A1 had the highest nAb titre (1:1024).

### 3.2. Neutralising Ab Titres over Time in Dutch IVIG

In the Dutch IVIG batches from 2010 and 2017, the nAb titres against types from *Enterovirus A*, *Enterovirus B* and *Parechovirus A* did not differ significantly between the two years (*p* < 0.05) (Figure 1b). Concerning the *Enterovirus A* species the nAb titres against CVA16 and EV-A71C1 increased > 2-fold over time (from 1:91 to 1:512 and from 1:54 to 1:152, respectively). The nAb titre against CVA6 tended to decrease. One *Enterovirus B* type, E30, showed an >2-fold increase between 2010 and 2017 (from 1:45 to 1:97). The highest nAb titre in 2010 and 2017 was found against CVA9 (1:1024 and 1:864, respectively). As for the Dutch 2017 batch, the included types from *Enterovirus C* were not neutralised by the Dutch 2010 batch. The titre against EV-D68 increased >2-fold over time. The nAb titre against this type increased from 1:41 in 2010 to 1:128 in 2017. The nAb titre against PeV-A3 increased >2-fold from 1:49 to 1:152.

### 3.3. Neutralising Ab Titres in 2010 Dutch and 2011 Vietnamese IVIG batch

The mean nAb titres against types from Enterovirus A, Enterovirus B and Parechovirus A did not differ significantly between the Dutch 2010 and Vietnamese 2011 IVIG batch (*p* < 0.05). An important difference in the Enterovirus A types was the >2-fold lower nAb titre against CVA6 (1:32 against 1:256, respectively) and >2-fold higher nAb titre against CVA16 (1:512 against 1:91, respectively) in the Vietnamese in comparison to the Dutch IVIG batch. No differences between the two batches for the different EV-A71 genetic clades C1 (Dutch strain) and C4 (Japanese strain) were observed. Neutralising Ab titres against the included Enterovirus B types were overall comparable between the two batches. Similar to the Dutch IVIG batches, the included Enterovirus C types were not neutralised by the Vietnamese batch. The nAb titres of the Vietnamese IVIG batch against PeV-A1, PeV-A3 and PeV-A4 were of the same size order as the Dutch IVIG batch.

## 4. Discussion

In our study, we used IVIG as a representative of the immunological profile of a population and investigated sero-prevalence of NPEV and PeV in IVIG batches from 2010, 2011 and 2017 from the Netherlands and Vietnam. 

The sero-prevalence in the Dutch IVIG batches was comparable to the VIRO-Typened geno-surveillance data of NPEV and PeV, with overall medium nAb titres present in IVIG against included *Enterovirus A*, *Enterovirus B* and *Parechovirus A* types. The GMT of included *Enterovirus B* species types was the highest, which corresponds to the fact that *Enterovirus B* members are the most commonly isolated types in VIRO-Typened [10]. No neutralisation of any included *Enterovirus C* type was observed. Types belonging to *Enterovirus C* species are rarely reported according to VIRO-Typened [10,15,16]. Our data suggest an absence of population immunity against these types. However, it is possible that the included *Enterovirus C* types, do not reflect circulating strains, as they were isolated in 1988. The Dutch IVIG batches not only neutralised the Dutch EV-A71C1 strain but also the Japanese EV-A71C4 strain. Indeed, in a study from van der Sanden et al., intra-typic cross-neutralisation was reported for some but not all EV-A71 genotypes (including EV-A71C1 and EV-A71C4) [6]. Another striking observation was the low nAb titre against EV-D68 in 2010, which increased in 2017. EV-D68 was rarely detected before 2010 when it started causing large outbreaks worldwide, the same year it had its first upsurge in VIRO-Typened [10,17]. In contrast, other research groups reported high nAb titres against a recent clinical EV-D68 B3 strain from 2016 in serum samples before and after the first reported upsurge in 2010 [18,19]. 

In contrast to our findings, a previous study from 1997 failed to show a correlation between nAb titres in two Dutch IVIG batches and the prevalence of certain serotypes. However, at that time surveillance was based on virus culture and serotyping, which is less sensitive than genotyping methods. Furthermore, mostly reference strains were used instead of current circulating strains [8].

Against some types we found unexpectedly low levels of nAb titres. While E30 is known to circulate endemically in the Netherlands and to cause large outbreaks in Vietnam [10,13,15,20,21], low nAb titres were found in all IVIG batches. It is possible, however, that other genetic clades are more dominant or of equally importance than the strains we included [22]. We found low nAb titres against CVA6 in the Vietnamese IVIG batch, while the last decade CVA6 has replaced CVA16 and EVA71 as the primary cause of HFMD in several Asian-Pasific countries [2]. It could be, however, that more recent IVIG batches from Vietnam would show higher levels of nAb titres against CVA6. Unfortunately, we did not have access to IVIG from Vietnam after 2011. Low nAb titres against PeV-A3 were detect in the 2010 Dutch IVIG batch. Other studies have shown high population seropositivity against PeV-A3 in the Netherlands but declining seropostivity in age groups over 30 could suggest that widespread circulation has emerged only recently [13,15,23]. In fact, nAb titres against PeV-A3 did increase in the 2017 Dutch IVIG batch. Low nAb titres against PeV-A3 were also found in the 2011 Vietnamese IVIG batch, where epidemics are known to occur every 2–3 years since 2006 [24]. Our study did not include a more recent Vietnamese IVIG batch, so we were unable to evaluate if nAb titres against PeV-A3 also increase over time in IVIG from Vietnam.

Our study had several strengths. We measured nAb titres against members from all NPEV and *Parechovirus A* species which gave a broad footprint. Furthermore, strains were selected to reflect as much as possible current circulating types. However, except for EV-A71, we used only a single clinical strain for each type. In general, it is assumed that cross-neutralisation is limited between different types [1]. Recent studies, however, did show differences in the level of intra-typic cross-neutralisation between different clades [6,25]. Therefore, we cannot differentiate between neutralisation and cross-neutralisation to other genetic (sub)types. As different genetic (sub)types can circulate and vary between regions and over time, it could explain some of the differences we found and appropriate strains have to be selected adapted to the time period and geographical region of the IVIG. Another limitation is that not all age groups were included in our study, as IVIG do not contain plasma from children. The incidence of EV and PeV infections is the highest in this age group and nAb measured in IVIG can be a reflection of infections encountered many years ago. This interval is important to consider when analysing changes over time. Finally, we cannot exclude an effect of different cell lines on nAb titres. In conclusion, we demonstrated that IVIG can be a valuable and reliable tool to generate sero-prevalence data. Moreover, it can be used to evaluate changes over time and to expose population susceptibility to certain (sub)types. Therefore, it could be an important tool, in addition to geno-surveillance data, in predicting future upsurges and outbreaks by upcoming geno-subtypes. Furthermore, it has the potential to provide data on circulating strains causing more mild disease. These types could be missed by geno-surveillance which is mostly based on sequence typing of EV and PeV found in clinical samples. As the *Enterovirus C* strains we studied, were isolated in 1988, more studies are needed to elucidate circulation and population immunity against *Enterovirus C* species. Finally, to be able to compare data generated by different sero-prevalence studies, reference strains and techniques should be further harmonised.

## Figures and Tables

**Figure 1 viruses-13-01028-f001:**
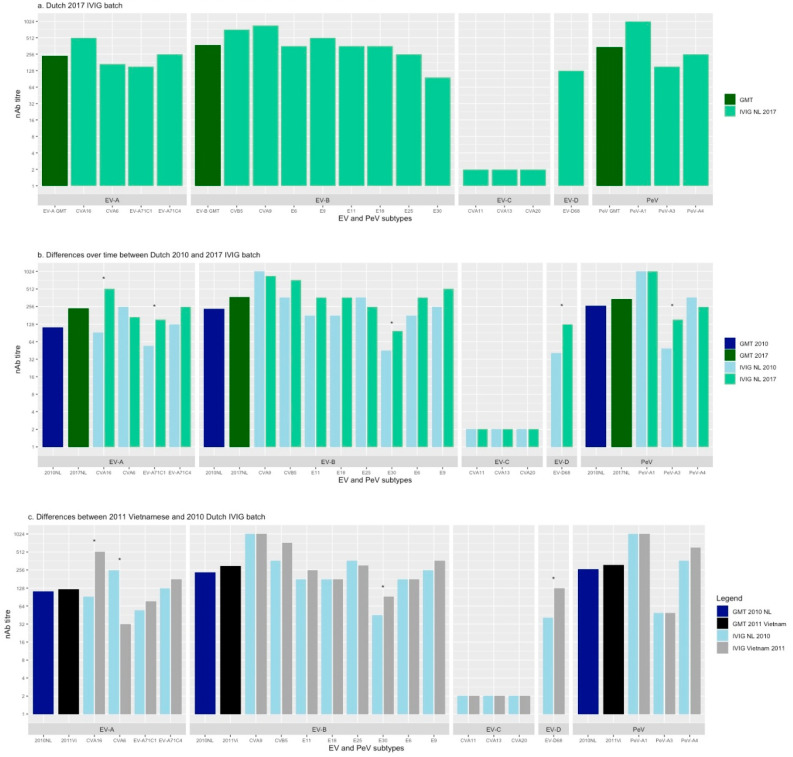
Neutralising antibody titres against different *Entero-* and *Parechovirus* strains **a**. Dutch 2017 IVIG batch **b**. Differences over time between Dutch 2010 and 2017 IVIG batch **c**. Differences between 2011 Vietnamese and 2010 Dutch IVIG batch. Asterisks (*) indicate a more than twofold difference in nAb titre between IVIG batches. GMT; geometric mean titre, IVIG; intravenous immunoglobulins, nAb titre; neutralising antibody titre, EV; enterovirus, PeV; parechovirus, CVA; coxsackievirus A, CVB; coxsackievirus B, E; echovirus, NL; the Netherlands, Vi; Vietnam.

**Table 1 viruses-13-01028-t001:** VIRO-Typened top five most prevalent viruses circulating between 2010 and 2017 in the Netherlands (adapted from Benschop 2016, source VIRO-TypeNed). EV; enterovirus, PeV; parechovirus, CVA; coxsackievirus A, CVB; coxsackievirus B, E; echovirus.

Ranking	2010	2011	2012	2013	2014	2015	2016	2017
**1**	CVA9	E-25	E-18	CVB3	E-16	E-11	E-30	E-5
**2**	EV-A71	E-7	CVA6	CVA9	E-25	CVA6	EV-D68	CVB5
**3**	PeV-A1	PeV-A1	PeV-A1	E-30	CVA6	E-9	E-6	CVA6
**4**	E-30	CVB3	PeV-A3	EV-A71	CVA16	E-18	CVB5	E-25
**5**	PeV-A3	CVB4	E-9	PeV-A1	EV-D68	CVB5	PeV-A3	E-9

## Data Availability

The data presented in this study are available in Appendix A.

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
