# Peer review of "Neutralising Antibodies against Enterovirus and Parechovirus in IVIG Reflect General Circulation: A Tool for Sero-Surveillance"

_viruses, 2021, doi:10.3390/v13061028_

Round 1

Reviewer 1 Report

I think the author’s challenge is quite interesting, although there are several limitations.   The authors evaluated intravenous immunoglobulins (IVIG) using Dutch batches in 2010 and 2017 and that from Vietnam in 2011 against a total of 20 enteroviruses (EVs) and parechoviruses (PeVs) to generate sero-surveillance data and compared the data with geno-surveillance data.   They suggested a good correlation with what is known from geno-survaillance data.   Especially, I was impressed that neutralising antibodies (nAbs) against strains belonging enterovirus C were not observed, whereas those against PeV-A1 were the highest.   My comments are follows.

1)Although the authors criticize that genotyping through VP1 sequencing of EV and PeV found in clinical samples is biased towards types causing disease and requiring hospitalization (lines 51-54), I think this approach is very important as a geno-surveillance for control of viral infectious diseases. Author’s proposal in this manuscript has another value and maybe surveillance among mild cases is also welcomed as the authors pointed out (lines 261-263).

2)Ideally, it might be better to compare sero-surveillance data using serum banks (a general sero-epidemiological survey) with those using IVIG (lines 60-64) as the first step.

3)It might be better to combine the explanation on selection of viruses (lines 133-147) with the section “Selection of NPEV and PeV strains (lines 82-95)”.

4)The nAbs against EV-D68 increased from 1:41 in 2010 to 1:128 in 2017 and those against PeV-A3 from 1:49 to 1:152 (lines 185-188). These findings are interesting and these data probably suggest that the infectious diseases related to these viruses increased recently.   However, the individual adults, who provided sera for IVIG in 2017, were affected by these viruses when they were still children or young.   Thus, we should be careful in time lag between the affected year and the collection year of their sera for IVIG.   As the author describe (line 256), the biggest weak point of this study is that IVIG do not contain sera from children, who are generally affected by picornaviruses.

5)The authors discussed on cross-neutralization between different types (lines 250-253). However, antigenic variation among the identical type is also suggested, as Karelehto E. et al reported for PeVA3 (Reference 11).   Is there antigenic variation among E30 strains?

Author Response

Dear reviewer,

Thank you for your valuable feedback and suggestions. It helped us to reflect more deeply on several elements of our study. We hope that our response will provide more clarity and that you find them satisfactory. You can find the changes made attached. 

  1. We do agree with the reviewer that geno-surveillance is also very important, therefore we rephrased this in the introduction (lines 52-53).

  2. Certainly, we find this a good suggestion. Including sero-banks in our study, however, was outside the scope of this study. Good sero-banks are rare and therefore valuable. And thus, we merely wanted to demonstrate the feasibility of using IVIG for sero-surveillance studies. However, we did compare our own results with existing geno- and also sero-surveillance data from the literature in our discussion (for example line 261 or line 278). This was not stated at the end of the introduction, therefore we specified this (line 69-70). However, these comparisons are challenging as often different strains, methods and years of sampling have been used. Furthermore, we are working on several sero-surveillance studies with serum samples. Whenever available we will include IVIG in these experiments as a comparison and follow-up of this study.

  3. We thank the reviewer for this suggestion and we combined the two sections under ‘Selection of NPEV and PeV’.

  4. Considering and reflecting on the time lag between the affected year and the collection year of blood for IVIG is a very important comment and indeed this was not discussed very clearly in the manuscript. We therefore dedicated some attention to it in our discussion (line 297-300).

  5. Yes, different geno-groups and clades have been described for E30 with amino-acid diversity in VP1. We added a reference to the part where we discuss the possibility of the influence of different genetic subtypes or clades on E30 neutralisation results (line 273).

Reviewer 2 Report

In the manuscript, DR Karen et al evaluated Intravenous immunoglobulins (IVIG) to generate sero-surveillance data. The authors performed neutralization test of nineteen EV and PeV (20 strains) with Dutch IVIG batches from 2 different time points (2010 and 2017) and an IVIG batch from Vietnam (2011). The results showed a good correlation with what is known from geno-surveillance data. The highest neutralization antibody titres were found against EV-B strains, while no neutralization antibody titres against EV-C strains. Thus, this study demonstrated that sero-surveillance by means of IVIG can be used to obtain insight into circulation of EV and PeV genotypes.

This is an interesting study, especially for public health, that can be used to evaluate changes over time and the susceptibility of populations to emerging genotypes.

The manuscript can be accepted after some minor comments for authors to consider.

  1, The authors realized that except for EV-A71 (C1 genotype and C4 genotype) , they used only a single clinical strain for each type (lines 249-250). In fact, the EV and PeV circulating in each country may belong to different genotypes, These viruses with different genotypes but belonging to the same serotype may have different cross neutralization ability. Therefore, the neutralization test results of IVIG in different countries to the same virus strain may be different (as showed in the figure 1). The authors need to explain this result in the discussion.

2, The author needs to compare the results of neutralization test with real serum and IVIG, so as to show that IVIG can replace real serum for neutralization test.

Author Response

Dear reviewer,

Thank you for your valuable feedback and suggestions. It helped us to reflect more deeply on several elements of our study. We hope that our response will provide more clarity and that you find them satisfactory. You can find the changes made attached. 

  1. This is an important point and we tried to discuss this briefly in the following section: “Recent studies however did show differences in the level of intra-typic cross-neutralisation between different clades [6,23]. We can therefore not differentiate between neutralisation and cross-neutralisation to other genetic (sub)types.” (line 292-293). We therefore added following comment: “As different genetic (sub)types can circulate and vary between regions and over time, it could explain some of the differences we found and appropriate strains have to be selected adapted to the time period and geographical region of the IVIG.” (line 293-296). However, differences between the 2 countries were in general minor.

  2. Certainly, we find this a good suggestion. Including sero-banks in our study, however, was outside the scope of this study. Good sero-banks are rare and therefore valuable. And thus, we merely wanted to demonstrate the feasibility of using IVIG for sero-surveillance studies. However, we did compare our own results with existing geno- and also sero-surveillance data from the literature in our discussion (for example line 261 or line 278). This was not stated at the end of the introduction, therefore we specified this (line 69-70). However, these comparisons are challenging as often different strains, methods and years of sampling have been used. Furthermore, we are working on several sero-surveillance studies with serum samples. Whenever available we will include IVIG in these experiments as a comparison and follow-up of this study.

Round 2

Reviewer 2 Report

The authors revised the manuscript according to the suggestions of the reviewers, and the quality of the revised version was improved. Overall the referee has no critical comments regarding of performance of the investigations. The experimental procedures are OK and the results are adequate for publication.